# Intradermal ChAdOx1 Vaccine Following Two CoronaVac Shots: A Case Report

**DOI:** 10.3390/vaccines9090990

**Published:** 2021-09-04

**Authors:** Ekachai Singhatiraj, Krit Pongpirul, Anan Jongkaewwattana, Nattiya Hirankarn

**Affiliations:** 1Department of Medicine, Bumrungrad International Hospital, Bangkok 10110, Thailand; ekachai.singhatiraj@gmail.com; 2Department of Preventive and Social Medicine, Faculty of Medicine, Chulalongkorn University, Bangkok 10330, Thailand; 3Department of International Health, Johns Hopkins Bloomberg School of Public Health, Baltimore, MD 21205, USA; 4National Center for Genetic Engineering and Biotechnology (BIOTEC), National Science and Technology Development Agency (NSTDA), Pathumthani 12120, Thailand; anan.jon@biotec.or.th; 5Center of Excellence in Immunology and Immune-Mediated Diseases, Department of Microbiology, Faculty of Medicine, Chulalongkorn University, Bangkok 10330, Thailand; nattiya.h@chula.ac.th

**Keywords:** COVID-19 vaccines, ChAdOx1-S, CoronaVac, intradermal injections, intramuscular injections

## Abstract

Inactivated SARS-CoV-2 vaccines are used in many countries with uncertain immunogenicity. Intradermal ChAdOx1 has been proposed as a resource-efficient heterologous third booster shot. A 52-year-old healthy male healthcare professional had received two intramuscular CoronaVac shots on 21 April and 23 May 2021, and volunteered to take a 0.1 mL ChAdOx1 vaccine intradermally on 29 June 2021, with minimal local reactions. The declining IgG levels against spike protein from the two CoronaVac shots increased to higher than 10,000 AU/mL two weeks after the intradermal ChAdOx1. Moreover, the neutralizing antibody increased from 66.77% to almost 100%. A ratio of 6.6:9.7 of IgA:IgG was observed. The 50% pseudovirus neutralization titer (PVNT50) against lentiviral pseudovirus bearing a codon-optimized spike gene (wild type, alpha, beta, and delta) were 1812.42, 822.99, 1025.42, 1347.13, respectively. The SARS-CoV-2-specific T cells to spike protein–peptide pools (532–788 SFU/10^6^ PBMCs) were detected. In conclusion, the antibody and cellular responses to the intradermal ChAdOx1, as a third booster dose in a healthy volunteer who received two intramuscular CoronaVac shots, revealed a dramatic increase in the total antibodies, including IgG, IgA, as well as T cell responses against spike protein. The immune response from intradermal ChAdOx1 should be further investigated in a larger population.

## 1. Introduction

Inactivated SARS-CoV-2 vaccines have been used worldwide, especially in low- and middle-income countries (LMICs), under emergency use. While the conclusive results from at least 16 phases I/phase II/phase III trials have not yet been available [1], the findings from a large observational study in Chile revealed that the effectiveness of CoronaVac (Sinovac Biotech, Beijing, China) after one course of two 0.5 mL intramuscular injections was 65.9% in preventing symptomatic disease, 87.5% for hospitalization, 90.3% for ICU admission, and 86.3% in reducing mortality [2]. The study findings suggested that a third booster might be needed. Nonetheless, it should be noted that the efficacy of CoronaVac in alleviating disease severity and reducing mortality has been inconsistent, mainly because of the variant composition in different countries.

A recent review reported a range of 60% to 94% efficacies of different COVID-19 vaccine platforms [3]. The currently available data suggested lower antibody responses to the inactivated virus and viral-vectored vaccines than to the mRNA and protein subunit vaccines [3]. Multiple approaches to achieve the goal of an effective vaccination are pragmatically important [3].

As an alternative for immune response enhancement, a heterologous prime-boost strategy has been proposed to elicit both humoral and cell-mediated immune responses that could lead to robust, broad, and long-lasting immunity. For instance, in the CombiVacS trial, healthy individuals who received ChAdOx1-S (Vaxzevria, Cambridge, AstraZeneca, UK), followed with the BNT162b2 (Comirnaty, BioNTech, Mainz, Germany), produced substantially higher levels of antibodies than the control group [4]. Nonetheless, more evidence is required to prove this speculation.

The fluctuating supplies of the various coronavirus vaccines increase the likelihood of the heterologous vaccination strategy and introduce a potential use of a fractional dosing scheme in mass vaccination campaigns in resource-limited settings [5]. As Thailand’s initial plan for the domestic manufacture and distribution of ChAdOx1 was delayed, CoronaVac has been given to most of the population, including healthcare professionals. Given the comparable immunogenicity between the low-dose intradermal and standard-dose intramuscular administration of the influenza vaccine [6], along with relatively fewer systematic side effects [5], intradermal ChAdOx1 has been proposed as a third heterologous booster.

## 2. Case Presentation

A 52-year-old healthy male healthcare professional had received two intramuscular CoronaVac shots on 21 April and 23 May 2021. He did not experience any notable side effects from the vaccination. He had never been infected with the SAR-CoV-2 virus. Before the vaccination, his blood tested negative for IgM and IgG by the SARS-CoV-2 antibody chemiluminescent microparticle immunoassay (CMIA).

His serum tested for the SARS-CoV-2 spike IgG antibody (Abbott Laboratories, IL, USA) on 17 June (57 and 25 days after the first and second doses, respectively), revealing a level of 853.6 AU/mL, whereas the SARS-CoV-2 surrogate virus neutralization test (cPass™, GenScript, NJ, USA) reported 66.77% (Figure 1). The IgG antibody decreased to 690.7 and 640.2 AU/mL on 1 and 6 July (39 and 44 days after the second dose, respectively), whereas the neutralization test reduced to 57.16% and 51.30%, respectively.

On 29 June he volunteered to take a 0.1 mL ChAdOx1 vaccine intradermally. Local reactions (redness, induration, and pain) at the injection site were observed for three days, with no systemic side effects.

Two weeks (12 July) after the intradermal ChAdOx1 booster, the IgG antibody increased to 10,465.20 AU/mL, and increased to 14,176.80 AU/mL at three weeks (19 July). Likewise, the neutralizing antibody increased to 99.58% and 99.66%, respectively.

His serum revealed a ratio of 6.6:9.7 of IgA:IgG, measured by using the anti-SARS-CoV-2 ELISA IgG and IgA kit, but negative for the anti-nucleocapsid IgG antibody (EUROIMMUN Medizinische Labordiagnostika AG, Lübeck, Germany). The 50% pseudovirus neutralization titer (PVNT50) against lentiviral pseudovirus bearing a codon-optimized spike gene (wild type, alpha, beta, and delta) were 1812.42, 822.99, 1025.42, and 1347.13, respectively [7].

In addition, his whole blood was tested for SARS-CoV-2-specific T cells, by interferon-γ enzyme-linked immune absorbent spot (ELISpot). ELISpot plates (Millipore, Merck, Darmstadt, Germany) were coated with human IFN-γ antibody (1-D1K, Mabtech; 5 μg/mL) overnight at 4 °C. Then, 250,000 PBMCs were seeded in each well and stimulated for 16 h with pools of spike and nucleocapsid SARS-CoV-2 peptides (2 μg/mL) (Genscript and BEI resources). Then, the ELISpot plate was developed with human biotinylated IFNγ detection antibodies (7-B6-1, Mabtech AB, Nacka Strand, Sweden; 1:2000), followed by incubation with streptavidin-AP (Mabtech AB, Nacka Strand, Sweden; 1:2000) and BCIP/NBT phosphatase substrate. Spot-forming units (SFU) were quantified with ImmunoSpot. To quantify positive peptide-specific responses, 2× mean spots of the unstimulated wells were subtracted from the pooled peptide-stimulated wells, and the results were expressed as SFU/10^6^ PBMCs. We excluded the results if the positive control wells—phytohemagglutinin (PHA) or cytomegalovirus (CMV) lysates—were negative. The SARS-CoV-2-specific T cells (788,532, 562 SFU/10^6^ PBMCs) were detected after stimulation with two pools of 316 peptides from the spike glycoprotein and a pool of 53 peptides from the receptor-binding domain with mutation N501Y, respectively (Figure 2). Only 18 SFU/10^6^ PBMCs were detected upon stimulation with a pool of nucleocapsid SARS-CoV-2 peptides.

## 3. Discussion

This case report helps prove that the intradermal injection of the viral vector vaccine, as a third heterologous booster, is safe, with a manageable local reaction and minimal systemic reaction. The healthy healthcare professional volunteer who received two standard-dose intramuscular CoronaVac shots revealed a dramatic increase in total antibodies, including IgG, IgA, as well as T cell responses against the spike protein after the low-dose intradermal ChAdOx1.

Given the diversity of vaccines against the SARS-CoV-2 virus, both humoral and cellular immune response data are essential for the comparative assessment of the efficacy and effectiveness of the vaccines that are offered in each country. The cellular immune response is crucial, if not more important, for preventing infection and minimizing the disease severity; however, the humoral immune response to a vaccine is relatively easier to measure and, therefore, has been common in the literature.

As vaccination with conventional vaccines in most cases results in the generation of a humoral, but not cellular, immune response, inactivated SARS-CoV-2 vaccines have been perceived as demonstrating a relatively lower efficacy than the other types of coronavirus vaccines. The waning of antibodies that are specific for SARS-CoV-2 spike protein has been reported in individuals vaccinated with CoronaVac [3,8]. However, evidence showing the level of antibodies at time points as early as 6 weeks, compared with 2 weeks after the second dose, is currently lacking. It should be noted, however, that the % inhibition, as assessed by the SARS-CoV-2 surrogate virus neutralization test in our study, may provide a number with a rather wide range of data. In our opinion, the numbers of 67% and 57.16% or 51.30% inhibition are subtle and are not considered statistically significant. Although evidence from robust head-to-head trials is not available, recent observational and epidemiological studies [2] concur with this hypothesis. It is not surprising that the Thai government has been heavily criticized when CoronaVac and BBIBP-CorV (Biological Institute of Biological Products, Beijing, China) [9] were ‘incidentally’ used in Thailand, while the initial plan for ChAdOx1 was not as anticipated and the other types of coronavirus vaccines—especially the mRNA vaccines—were not available. Of the 16 million shots given, more than half were inactivated SARS-CoV-2 vaccines (CoronaVac 50.85% and BBIBP-CorV 4.42%), whereas 44.73% were ChAdOx1 [10]. Nonetheless, the situation gave us an opportunity to explore the use of the viral vector vaccine as a third booster dose among those who already received at least one shot of the inactivated SARS-CoV-2 vaccine.

As a viral vector vaccine, ChAdOx1 has been widely used, and several trials have been ongoing, but no evidence on the intradermal administration of viral vector vaccines exists. Only three human studies have demonstrated an excellent humoral and cellular immune response to the intradermal administration of plasmid DNA—ZyCoV-D (Zydus Cadila, Ahmedabad, India) [11], INO-4800 (Inovio Pharmaceuticals, Plymouth Meeting, PA, USA) [12]—and mRNA-1273 (Moderna, Cambridge, MA, USA) [13] vaccines. While the intradermal route was used in the animal models of both the DNA vaccines [14,15], the development of conventional inactivated SARS-CoV-2 or viral vector vaccines has relied only upon intramuscular administration.

The neutralizing antibody profile reported in this work was quite distinct from what others have reported by using the pseudotyped-based system. In fact, we usually observed lower responses when testing most of the sera from vaccinated individuals with the pseudovirus bearing the beta variant’s spike. However, this specific case is rather exceptional, as the neutralizing activity against the beta pseudovirus appeared to be comparable to other VOCs. To our knowledge, this pattern of response is indeed observed in a so-called “elite responder”, whose neutralizing activity against the beta variant of SARS-CoV-2 could be as high as other VOCs [16]. We speculate, based on the high antibody titer after ID boosting, that the subject in our study may fall into the category of the elite responder. More data are needed, from more subjects, to verify that this observed pattern is common among those boosted by the ID strategy.

While intradermal administration might seem difficult for inexperienced staff, the side effects from intradermal injection were mostly local, whereas the potential complications were not as severe as the thrombosis with thrombocytopenia syndrome (TTS) from accidental intravenous ChAdOx1 injection [17]. Intradermal vaccination against the SARS-CoV-2 virus could be safely and economically offered en masse, especially with more-advanced instruments, such as jet injection, transdermal patches, and micro-needles [17,18].

Nonetheless, given that there were no homologous or intramuscular controls, this case report could not directly compare between different boosting regimens or routes of administration.

## 4. Conclusions

Antibody and cellular responses to the intradermal ChAdOx1 vaccination, as a third booster dose in a healthy volunteer who received two intramuscular CoronaVac shots, revealed a dramatic increase in the total antibodies, including IgG, IgA, as well as T cell responses against the spike protein. The immune response from a low-dose intradermal ChAdOx1 should be further investigated in a larger population.

## Figures and Tables

**Figure 1 vaccines-09-00990-f001:**
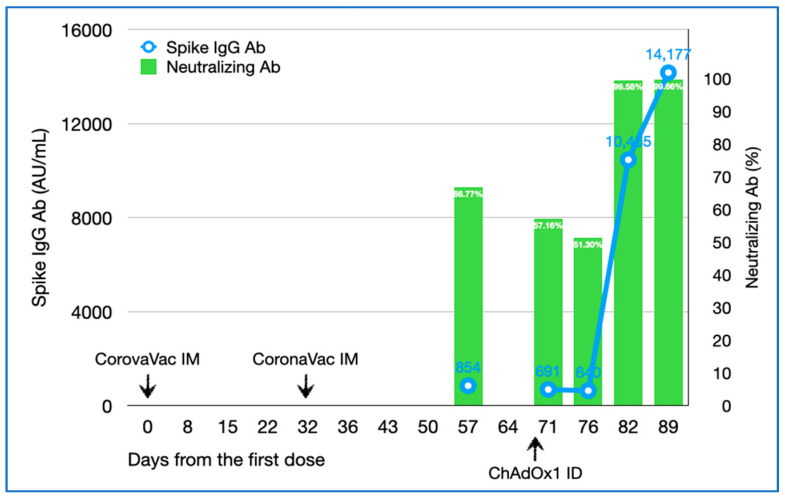
Spike immunoglobulin G and neutralizing antibody levels after the two intramuscular CoronaVac vaccinations and one intradermal ChAdOx1 vaccination. The green bar graphs present neutralizing antibody levels (%) whereas the blue line graph presents spike IgG levels (AU/mL). Ab, antibody; ID, intradermal; IM, intramuscular; IgG, immunoglobulin G.

**Figure 2 vaccines-09-00990-f002:**
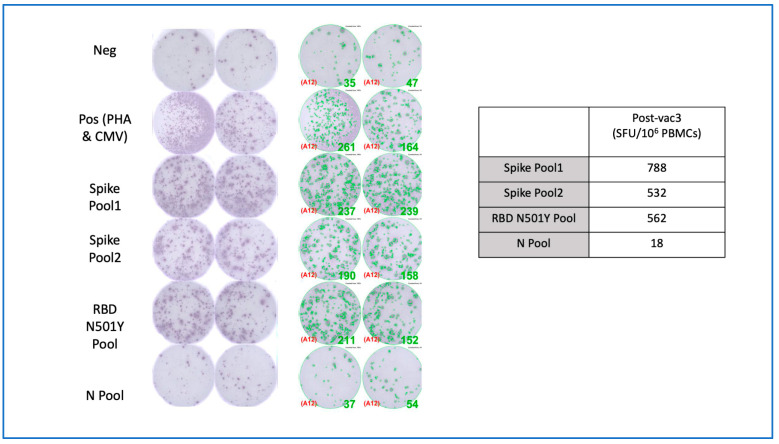
SARS-CoV-2-specific T cells were detected using interferon-gamma ELISpot assay after the intradermal ChAdOx1. The figure showed scanned images and SFU/well counts for unstimulated and stimulated PBMCs. The final SFU per one million PBMCs after subtracting the background was summarized in a table. ELISpot, enzyme-linked Immunospot; Neg, negative; Pos, positive; PHA, phytohemagglutinin; CMV, cytomegalovirus; RBD, receptor-binding domain; SFU, spot-forming unit; PBMCs, peripheral blood mononuclear cells.

## Data Availability

Not applicable.

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
