# Peer review of "Intradermal ChAdOx1 Vaccine Following Two CoronaVac Shots: A Case Report"

_vaccines, 2021, doi:10.3390/vaccines9090990_

Round 1

Reviewer 1 Report

Vaccination is the only effective method for controlling the COVID-19 pandemic; however, a lack of vaccine supply, particularly in low-income countries greatly delay COVID-19 vaccination. In addition, the immune response in vaccinated persons is decaying as time goes on, and they may need further boosters. In the manuscript entitled “ Intradermal ChAdOx1 Vaccine following Two CoronaVac Shots: A Case Report”, Singhatiraj et al described the a case of ChAdOx1 intradermal boosting with an individual who had received two doses of CoronaVac vaccination. The results showed that the intradermal boosting not only elicited much higher antibody titers, but also induced T cell response against spike protein, which is consistent with previous study that a heterologous prime-boost method can elicit much higher levels of antibodies than the no-boost group. This finding proved that the third dose booting could significantly increase the immune response against COVID-19 infection particularly in people received two doses of CoronaVac.

The study was well planned and carried out, although no definitive conclusions can be taken from this single case, it does provide an interesting avenue for COVID vaccine administration. The new protocol is helpful not just for boosting but also for overcoming the vaccine shortage. However, because there was no homologous boosting control and no intramuscular boosting control, the study was unable to answer some of the important problems, such as whether heterologous boosting can induce a greater immune response than homologous boosting; and if intradermal boosting is a better option than intramuscular boosting. In addition, I still have a few concerns that need to be addressed.

  1. Lines 51 to 57, the author described, “As a better alternative for immune response enhancement, a heterologous prime-boost strategy has been proposed to elicit both humoral and cell-mediated immune responses resulting in more robust, broader, or longer-lasting immunity.” Furthermore, reference 3 was cited as an example. However, in the results of reference 3 only proved that boosting with BNT162b2 can elicit a robust immune response when compared to no boosting group, but no evidence showed that the heterologous boosting is better than homologous boosting in inducing immune response against COVID-19. Therefore, please cite the right evidence(s) that support your claim, otherwise, please revise the manuscript accordingly.
  2. The title of Figure 1 is not accurate, additional information is needed to describe the meaning of numbers and percentages in the figure.
  3. The title of Figure 2 should also be altered to more accurately represent the information in the figure. A legend explaining the images and table should also be included.

It would be preferable for future research to include an intramuscular boosting with CoronaVac group as a control group in addition to the no boosting control group.

Author Response

Vaccination is the only effective method for controlling the COVID-19 pandemic; however, a lack of vaccine supply, particularly in low-income countries greatly delay COVID-19 vaccination. In addition, the immune response in vaccinated persons is decaying as time goes on, and they may need further boosters. In the manuscript entitled “ Intradermal ChAdOx1 Vaccine following Two CoronaVac Shots: A Case Report”, Singhatiraj et al described the a case of ChAdOx1 intradermal boosting with an individual who had received two doses of CoronaVac vaccination. The results showed that the intradermal boosting not only elicited much higher antibody titers, but also induced T cell response against spike protein, which is consistent with previous study that a heterologous prime-boost method can elicit much higher levels of antibodies than the no-boost group. This finding proved that the third dose booting could significantly increase the immune response against COVID-19 infection particularly in people received two doses of CoronaVac.

Response: Thank you very much for the comments that precisely summarized our efforts.

The study was well planned and carried out, although no definitive conclusions can be taken from this single case, it does provide an interesting avenue for COVID vaccine administration. The new protocol is helpful not just for boosting but also for overcoming the vaccine shortage. However, because there was no homologous boosting control and no intramuscular boosting control, the study was unable to answer some of the important problems, such as whether heterologous boosting can induce a greater immune response than homologous boosting; and if intradermal boosting is a better option than intramuscular boosting. In addition, I still have a few concerns that need to be addressed.

Response: Thank you very much for the compliments. Yes, this case report had no homologous or intramuscular controls and, therefore, could not offer a definitive conclusion. This point was added as a limitation in the Discussion section.

“Nonetheless, given no homologous or intramuscular controls, this case report could not directly compare between different boosting regimens or routes of administration.”

  1. Lines 51 to 57, the author described, “As a better alternative for immune response enhancement, a heterologous prime-boost strategy has been proposed to elicit both humoral and cell-mediated immune responses resulting in more robust, broader, or longer-lasting immunity.” Furthermore, reference 3 was cited as an example. However, in the results of reference 3 only proved that boosting with BNT162b2 can elicit a robust immune response when compared to no boosting group, but no evidence showed that the heterologous boosting is better than homologous boosting in inducing immune response against COVID-19. Therefore, please cite the right evidence(s) that support your claim, otherwise, please revise the manuscript accordingly.

Response: Thank you very much for pointing out this important concern. We agree with your advice and toned down the statements in the revised manuscript accordingly.

“As an alternative for immune response enhancement, a heterologous prime-boost strategy has been proposed to elicit both humoral and cell-mediated immune responses that could lead to a robust, broad, and long-lasting immunity. … Nonetheless, more evidence is required to prove this speculation.”

  1. The title of Figure 1 is not accurate, additional information is needed to describe the meaning of numbers and percentages in the figure.

Response: Thank you very much. The title of Figure 1 was changed to “Figure 1 Spike Immunoglobulin G and Neutralizing Antibody Levels after the Two Intramuscular CoronaVac and One Intradermal ChAdOx1 Vaccination.” The legend for Figure 1 was added: “The green bar graphs present neutralizing antibody levels (%) whereas the blue line graph presents Spike IgG levels (AU/mL). Ab, Antibody; ID, intradermal; IM, intramuscular; IgG, Immunoglobulin G. Ab, Antibody; ID, intradermal; IM, intramuscular; IgG, Immunoglobulin G.”

  1. The title of Figure 2 should also be altered to more accurately represent the information in the figure. A legend explaining the images and table should also be included.

Response: The title and legend for Figure 2 were revised as follows: “Figure 2 SARS-CoV-2-specific T Cells were detected using interferon-gamma ELISpot assay after the intradermal ChAdOx1. The figure showed scanned images and SFU/well counts for unstimulated and stimulated PBMCs. The final SFU per 1 million PBMCs after subtracting the background was summarized in a table. ELISpot, En-zyme-linked Immunospot; Neg, Negative; Pos, Positive; PHA, Phytohe-magglutinin; CMV, Cytomegalovirus; RBD, Receptor Binding Domain; SFU, Spot Forming Unit; PBMCs, Peripheral Blood Mononuclear Cells.”

It would be preferable for future research to include an intramuscular boosting with CoronaVac group as a control group in addition to the no boosting control group.

Response: Thank you very much. Your advice is well taken into the planning of our research proposal.

Reviewer 2 Report

The paper presents the results of a case study where ChAdOx1 vaccine was given as heterologous third dose in a person vaccinated with two doses of CoronaVac vaccine.

The text of the report is correct.

Figure 1 presenting the results must be improved. It is necesary to indicate marks for levels on x and y axis. The values of neutralizing Ab on bars are not clear.

Author Response

The paper presents the results of a case study where ChAdOx1 vaccine was given as heterologous third dose in a person vaccinated with two doses of CoronaVac vaccine. The text of the report is correct.

Response: Thank you very much for your compliment.

Figure 1 presenting the results must be improved. It is necessary to indicate marks for levels on x and y axis. The values of neutralizing Ab on bars are not clear.

Response: Thank you very much. The title of Figure 1 was changed to “Figure 1 Spike Immunoglobulin G and Neutralizing Antibody Levels after the Two Intramuscular CoronaVac and One Intradermal ChAdOx1 Vaccination.” The legend for Figure 1 was added: “The green bar graphs present neutralizing antibody levels (%) whereas the blue line graph presents Spike IgG levels (AU/mL). Ab, Antibody; ID, intradermal; IM, intramuscular; IgG, Immunoglobulin G. Ab, Antibody; ID, intradermal; IM, intramuscular; IgG, Immunoglobulin G.”

Reviewer 3 Report

Summary: The study addresses the benefit of a third booster shot. This is a case report of an individual that was vaccinated with two standard CoronaVac shots and a month later, received a single intradermal shot of ChAdOx1. A follow up on the resulted immunity (IgG, IgA T cell) against spike protein is reported.

Major comments:

  1. Reports on CoronaVac are based on observational studies that are very noisy and are complicated due to changes in variant composition in different countries. The details “65.9% in preventing symptomatic disease……and 86.3% in reducing mortality” should come with a disclaimer. The WHO reports on CoronaVac- “A large phase 3 trial in Brazil showed that two doses, administered at an interval of 14 days, had an efficacy of 51% against symptomatic…” . From observational in Brazil other numbers are shared: “Estimated vaccine effectiveness of 49.6% following at least one dose and 50.7% two weeks after the second dose for the P.2 variant”. Make sure to reflect the inconsistency and the strong dependency on the circulating variants.
  2. Is there evidence showing that with CoronaVac protection drops in IgG / neutralization within such short time? (39 to 44 days from 2nd shot). The drop from 67% to 51% in neutralization test is surprisingly fast. Please add relevant data.
  3. Results from wt, Alpha, Beta, and Delta were 1812.42, 822.99, 1025.42, 1347.13, respectively. Those results are very different from the reported identical assays of Pfeizer/ Moderna results. Here, it is suggested (based on a single case) that a weak dependency is shown for major VOC. Is there external evidence supporting this observation?
  4. It is of course impossible to generalize from a single case. Therefore, it is important to report whether this case (prior to the 3rd shot) matches average level of IgG (by age, gender and time of vaccination). Such information is essential to get a feeling for the studied case.
  5. The claim that increased immunity is attributed to the intradermal shot is not validated. It is most likely caused by the benefit of the boost itself. Recent reports from Israel in which ~1.8M already received a (muscle injection) 3rd boost showed a dramatic elevation in IgG within days. Therefore, limitations of interpretation should be better stated.
  6. Minor comments:

    1. Remove the structured abstract.
    2. The legend for Fig 1 should be added (include abbreviation of IM)
    3. The use of 3 digits is not needed, better to say 6:9.7 for IgA:IgG. Add the level of confidence on these numbers (in case it was repeatedly done)
    4. Add a review on different vaccine effectiveness (Nature Reviews Immunology 21, pg 475–484; 2021).

Author Response

Summary: The study addresses the benefit of a third booster shot. This is a case report of an individual that was vaccinated with two standard CoronaVac shots and a month later, received a single intradermal shot of ChAdOx1. A follow up on the resulted immunity (IgG, IgA T cell) against spike protein is reported.

Major comments:

  1. Reports on CoronaVac are based on observational studies that are very noisy and are complicated due to changes in variant composition in different countries. The details “65.9% in preventing symptomatic disease……and 86.3% in reducing mortality” should come with a disclaimer. The WHO reports on CoronaVac- “A large phase 3 trial in Brazil showed that two doses, administered at an interval of 14 days, had an efficacy of 51% against symptomatic…”. From observational in Brazil other numbers are shared: “Estimated vaccine effectiveness of 49.6% following at least one dose and 50.7% two weeks after the second dose for the P.2 variant”. Make sure to reflect the inconsistency and the strong dependency on the circulating variants.

Response: Thank you very much. We added a disclaimer to the first paragraph of the Introduction section as suggested.

“Nonetheless, it should be noted that the efficacy of CoronaVac in alleviating disease severity and reducing mortality has been inconsistent, mainly because of the variant composition in different countries.”

  1. Is there evidence showing that with CoronaVac protection drops in IgG / neutralization within such short time? (39 to 44 days from 2ndshot). The drop from 67% to 51% in neutralization test is surprisingly fast. Please add relevant data.

Response: Thank you for the comment. The waning of antibodies specific for SARS-CoV-2 spike has been reported in individuals vaccinated with CoronaVac (Souza et al. Neutralisation of SARS-CoV-2 lineage P.1 by antibodies elicited through natural SARS-CoV-2 infection or vaccination with an inactivated SARS-CoV-2 vaccine: an immunological study. Lancet Microbe 2021 and Zhang et al. Studies on the level of neutralizing antibodies produced by inactivated COVID-19 vaccines in the real world. medRxiv 2021). However, the evidence showing the level of antibodies at time points as early as 6 weeks compared with 2 weeks after the second dose is currently lacking. It should be noted, however, that the % inhibition as assessed by the SARS-CoV-2 Surrogate Virus Neutralization Test in our study may provide a number with a rather wide range of data. In our opinion, the numbers of 67% and 57.16 or 51.30% inhibition are subtle and considered not statistically significant. These statements were added to the Discussion section.

  1. Results from wt, Alpha, Beta, and Delta were 1812.42, 822.99, 1025.42, 1347.13, respectively. Those results are very different from the reported identical assays of Pfeizer/Moderna results. Here, it is suggested (based on a single case) that a weak dependency is shown for major VOC. Is there external evidence supporting this observation?

Response: We agree with the reviewer that the neutralizing antibody profile reported in this work is quite distinct from what others have reported using the pseudotyped-based system. We usually observed lower responses when testing most of the sera from vaccinated individuals with the pseudovirus bearing the Beta variant's spike. However, this specific case is rather exceptional as the neutralizing activity against the Beta pseudovirus appeared comparable to other VOCs. To our knowledge, this pattern of response is indeed observed in a so-called “Elite Responder” whose neutralizing activity against the Beta variant of SARS-CoV-2 could be as high as other VOCs (Maeda et al. Correlates of Neutralizing/SARS-CoV-2-S1-binding Antibody Response with Adverse Effects and Immune Kinetics in BNT162b2-Vaccinated Individuals. medRxiv 2021). We speculate, based on the high antibody titer after ID boosting, that the subject in our study may fall into the category of the Elite Responder. More data are needed from more subjects to verify that this observed pattern is common among those boosted by the ID strategy. These statements were added to the Discussion section.

  1. It is of course impossible to generalize from a single case. Therefore, it is important to report whether this case (prior to the 3rdshot) matches average level of IgG (by age, gender and time of vaccination). Such information is essential to get a feeling for the studied case.

Response: Thank you very much for this important point. We were able to retrieve the Covid-19 Spike IgG antibody data from the hospital laboratory information system (LIS). Of 48 Thai male individuals who received two CoronaVac shots in this LIS dataset, six males aged 51-55 years. The median IgG level was 350.90 (IQR 198.80-4723.40) AU/mL whereas the mean was 3,122.40 (SD 5,132.58) AU/mL. Although we agree that this data could be useful for the readers but the formal analysis of this LIS data will be performed once the dedicated proposal is approved by the ethics committee shortly. Hence, we propose that these preliminary data are presented only as part of the response to the reviewer’s comment here but not in the manuscript.

  1. The claim that increased immunity is attributed to the intradermal shot is not validated. It is most likely caused by the benefit of the boost itself. Recent reports from Israel in which ~1.8M already received a (muscle injection) 3rdboost showed a dramatic elevation in IgG within days. Therefore, limitations of interpretation should be better stated.

Response: This case report had no homologous or intramuscular controls and, therefore, could not offer a definitive conclusion. This point was added as a limitation in the Discussion section.

“Nonetheless, given no homologous or intramuscular controls, this case report could not directly compare  between different boosting regimens or routes of administration.”

Minor comments: 

  1. Remove the structured abstract.

Response: The abstract was revised to be unstructured as suggested.

  1. The legend for Fig 1 should be added (include abbreviation of IM)

Response: The legend for Figure 1 was added: “The green bar graphs present neutralizing antibody levels (%) whereas the blue line graph presents Spike IgG levels (AU/mL). Ab, Antibody; ID, intradermal; IM, intramuscular; IgG, Immunoglobulin G. Ab, Antibody; ID, intradermal; IM, intramuscular; IgG, Immunoglobulin G.” Also, the title of Figure 1 was changed to “Figure 1 Spike Immunoglobulin G and Neutralizing Antibody Levels after the Two Intramuscular CoronaVac and One Intradermal ChAdOx1 Vaccination.”

  1. The use of 3 digits is not needed, better to say 6.6:9.7 for IgA:IgG. Add the level of confidence on these numbers (in case it was repeatedly done)

Response: Thank you for the advice. The digit was changed to be single because it was performed once.

  1. Add a review on different vaccine effectiveness (Nature Reviews Immunology 21, pg 475–484; 2021).

Response: Thank you very much. The suggested reference was added to the Introduction section.

A recent review reported a range of 60% to 94% efficacies of different COVID-19 vaccine platforms. Currently available data suggested lower antibody responses to the inactivated virus and viral-vectored vaccines than to the mRNA and protein subunit vaccines.”

Round 2

Reviewer 2 Report

the revised version of the paper has i mproved the original one.

The title of the figure clarify the results presented in the figure. 

Reviewer 3 Report

The authors replied to all important issues in a satisfactory way. 

Minor:

Please define VOC the first timeused